 

META-RESEARCH

# Investigating disagreement in the scientific literature

**Abstract** Disagreement is essential to scientific progress but the extent of disagreement in science, its evolution over time, and the fields in which it happens remain poorly understood. Here we report the development of an approach based on cue phrases that can identify instances of disagreement in scientific articles. These instances are sentences in an article that cite other articles. Applying this approach to a collection of more than four million English-language articles published between 2000 and 2015 period, we determine the level of disagreement in five broad fields within the scientific literature (biomedical and health sciences; life and earth sciences; mathematics and computer science; physical sciences and engineering; and social sciences and humanities) and 817 meso-level fields. Overall, the level of disagreement is highest in the social sciences and humanities, and lowest in mathematics and computer science. However, there is considerable heterogeneity across the meso-level fields, revealing the importance of local disciplinary cultures and the epistemic characteristics of disagreement. Analysis at the level of individual articles reveals notable episodes of disagreement in science, and illustrates how methodological artifacts can confound analyses of scientific texts.

**WOUT S LAMERS\*, KEVIN BOYACK, VINCENT LARIVIÈRE, CASSIDY R SUGIMOTO, NEES JAN VAN ECK, LUDO WALTMAN AND DAKOTA MURRAY\***

**\*For correspondence:**
w.s.lamers@cwts.leidenuniv.nl
(WSL);
dakmurra@iu.edu (DM)

**Competing interest:** The authors declare that no competing interests exist.

## Introduction

Disagreement is a common phenomenon in science, and many of the most famous discoveries in the history of science were accompanied by controversy and disputes. Dialectic discourse emerged in ancient Greece, whereby the truth was thought to emerge from the arguments and counterarguments of scholars engaged in dialogue. The modern scientific method arose from a similar dialogue 350 years ago, as two individuals—Robert Boyle and Thomas Hobbes—debated over the meaning of experimental results obtained with the newly-invented air pump (*Shapin and Schaffer, 2011*).

Disagreement also anchors much of the lore surrounding major scientific discoveries. For example, Alfred Wegener's theory of plate tectonics was initially rejected by the scientific community; it took decades for the existence of gravitational waves to be confirmed in physics (*Collins, 2017*) and the value of the Hubble constant is still disputed in cosmology (*Castelvecchi, 2020*). Other conflicts are influenced by forces external to science, such as the controversies on the link between cigarette and lung cancer or between greenhouse gas and climate change (*Oreskes and Conway, 2011*). Disagreement also features prominently in a number of influential theories in the philosophy and sociology of science, such as falsifiability (*Popper and Hudson, 1963*), paradigm shifts (*Kuhn, 1996*), and the scientific division of labor (*Kitcher, 1995*).

Despite its importance to science, however, there is little empirical evidence of how much disagreement exists, where it is most common, and its consequences. Quantitative measures can be valuable tools to better understand the role and extent of disagreement across fields of science. Previous research has focused on consensus as evidenced by citation networks (*Bruggeman et al., 2012*; *Shwed and Bearman, 2010*; *Shwed and Bearman, 2012*); on concepts related to disagreement in scientific texts such as negative citations, disputing citations, and uncertainty (*Catalini et al., 2015*; *Chen et al., 2018*; *Nicholson et al., 2021*); and on approaches based on word counts (*Bertin et al., 2016*). Studying disagreement is challenging, given the lack of a widely accepted theoretical framework for conceptualizing disagreement combined with major challenges in its operationalization, for

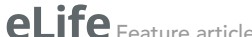

**Figure 1.** Agreement and validity of different combinations of signal term and filter term. Measures calculated from 50 randomly-sampled citances for each combination of signal term (vertical axis) and filter term (horizontal axis), annotated as valid or invalid instances of disagreement by two independent coders. (a) Percentage agreement, or the proportion of citances for which coders independently agreed on the label. (b) Percentage validity, or the proportion of citances which both coders labeled as valid. Averages for the various signal terms are shown in the left-most column; averages for the various filter terms are shown in the bottom row. (c) Percentage agreement (blue circles) and validity (red diamonds) of each signal/filter term combination, ordered from highest percent validity (top) to lowest percent validity (bottom). Numbers on the right are the total number of citances returned by querying using the signal/filter term combination, and are colored according to their log-transformed value. (d) Log-transformed count of citances returned by each query combination, colored by the (log-transformed) number of citances. Citance counts are non-exclusive, meaning that citances of the form *debat* + studies* will also be counted towards *debat* _standalone_*.

The online version of this article includes the following figure supplement(s) for figure 1:

**Figure supplement 1.** Distribution of citances returned by signal/filter term queries.

**Table 1.** Specific terms comprising each of the thirteen signal term sets and specific exceptions. The "*" symbol (wildcard) captures possible variants.

| Signal term | Variants | Exclusions | Results |
|---|---|---|---|
| challenge* | | | 405,613 |
| conflict* | | | 212,246 |
| contradict* | | | 115,375 |
| contrary | | | 171,711 |
| contrast* | | | 1,257,866 |
| controvers* | | | 154,608 |
| debat* | | "parliament* debat*", "congress* debat*", "senate* debat*", "polic* debat*", "politic* debat*", "public* debat*", "societ* debat*" | 150,617 |
| differ* | | "different*" | 2,003,677 |
| disagree* | "not agree*", "no agreement" | "range", "scale", "kappa", "likert", "agree*" and/or "disagree" within a ten-word range of each other. | 52,615 |
| disprov* | | "prove*" and "disprove*" within a ten-word range | 2,938 |
| no consensus | "lack of consensus" | "consensus sequence", "consensus site" | 16,632 |
| questionable | | | 24,244 |
| refut* | | "refutab*" | 10,322 |
| total | | | 4,578,464 |

instance, the limited availability of large-scale collections of scientific texts.

This paper proposes an operationalization of disagreement in scientific articles that captures direct disagreement between two papers, as well as statements indicative of disagreement within the community. We describe a methodological approach to generate and manually-validate cue-phrases that reliably match to citation sentences (which we call "citances") to represent valid instances of disagreement. We then use this approach to quantify the extent of disagreement across more than four million publications in the Elsevier *ScienceDirect* database, and investigate the rate of disagreement across fields of science.

## Literature review

It is widely acknowledged that disagreement plays a fundamental role in scientific progress (*Balietti et al., 2015*; *Sarewitz, 2011*; *Nature Methods, 2016*). However, few studies have tried to quantify the level of disagreement in the scientific literature. Part of this may be explained by the fact that disagreement is difficult to both define and measure. There have been, however, attempts to assess consensus or uncertainty in the literature. Much of the early work on consensus attempted at characterizing differences between so-called hard and soft sciences. Cole described a series of experiments done in several fields, finding no evidence of differences in cognitive consensus along the "hierarchy of sciences" (*Cole, 1983*). Hargens claimed that consensus was lower in fields having journals with higher rejection rates (*Hargens, 1988*). This claim was contested by Cole, Simon and Cole, who argued that other variables accounted for the differences, and that reviewer's assessments would be a better measure of consensus than rejection rates (*Cole et al., 1988*). Fanelli found that positive results—support for the paper's hypotheses—was far higher in the social sciences than the physical sciences, which is argued to reflect higher ambiguity, and thus lower consensus, in the social sciences (*Fanelli, 2010*).

Recent studies on scientific consensus have made use of citations and text. Through a series of case studies, Shwed and Bearman used

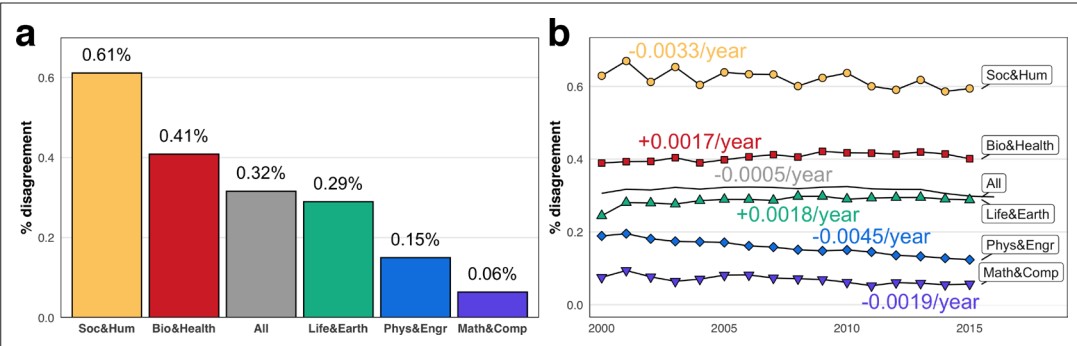

**Figure 2.** Disagreement reflects a hierarchy of fields. (**a**) Percent of all citances in each field that contain signals of disagreement, meaning they were returned by one of the 23 queries with validity of 80% or higher. Fields marked by lower consensus, such as in Soc & Hum, had a greater proportion of disagreement. (**b**) Percent of disagreement by field and over time, showing little change overall, but some changes by field. Text indicates the average percentage-point change per-year by field.

The online version of this article includes the following figure supplement(s) for figure 2:

**Figure supplement 1.** Percent of all citances returned by each of the 23 queries with validity over 80%.

network modularity to claim that divisions in the citation network decreased over time, corresponding to increased consensus (**Shwed and Bearman, 2010**). Nicolaisen and Frandsen used a Gini index calculated over bibliographic coupling count distributions to approximate consensus, and found that physics papers showed more consensus on average than psychology papers (**Nicolaisen and Frandsen, 2012**). Using a corpus of nearly 168,000 papers, Evans, Gomez and McFarland calculated the Shannon entropy of language in a set of eight fields, and found more evidence that consensus was higher in the hard sciences than the social sciences (**Evans et al., 2016**).

Other studies developed methods to identify *uncertainty* in text, a concept that is related to disagreement, and a potential indicator of consensus. For example, Szarvas and colleagues interpreted uncertainty as a "lack of information" and created a cue-word based uncertainty detection model based on three annotated datasets: BioScope, WikiWeasel, and Fact Bank (**Szarvas et al., 2012**). Their results suggest that while domain-specific cues are useful, there remain cues that can reasonably identify uncertainty across domains. Similarly, Yang and colleagues developed a classifier based on manually annotated uncertainty cues and conditional random fields, and conducted a series of experiments to assess the performance of their method (**Yang et al., 2012**). Chen, Song and Heo later extended these approaches and applied them to an empirical study of uncertainty across science (**Chen et al., 2018**). They first introduced a conceptual

framework to study uncertainty that incorporates epistemic status and perturbation strength, and then measured uncertainty in 24 high-level scientific fields, and finally created an expanded set of uncertainty cues to support further analysis (Note that these rates included all types of uncertainty, whether they be theoretical, conceptual, or experimental, and within or between studies). The reported rate of uncertainty closely mirrored consensus, highest in the social sciences, followed by the medical sciences, environmental sciences, and engineering.

Many of the cues used as a starting point by **Chen et al., 2018** are hedging terms, which are commonly used in scientific writing to express possibility rather than certainty (**Hyland, 1998**). In addition to being field-dependent, hedging rates have also been found to depend on whether a paper is primarily methodological. Recent work by Small and colleagues showed that citing sentences (i.e., citances) with the word "may" occur more frequently when citing method papers than non-method papers (**Small, 2018**; **Small et al., 2019**). More recently, Bornmann, Wray and Haunschild used a similar method to investigate uncertainty associated with specific concepts in the context of highly cited works (**Bornmann et al., 2019**). While some might equate uncertainty or hedging with disagreement, they are not the same. As Small and colleagues have written, when citing another work, "hedging does not assert that the paper is wrong, but only suggests that uncertainty surrounds some aspect of the ideas put forward" (**Small et al., 2019**). Here, we attempt to explicitly identify and measure

**Table 2.** Specific terms comprising each of the four filter term sets.

| studies | studies; study; previous work; earlier work; literature; analysis; analyses; report; reports |
| --- | --- |
| ideas | idea*; theory; theories; assumption*; hypothesis; hypotheses |
| methods | model*, method*, approach*; technique* |
| results | result*; finding*; outcome*; evidence; data; conclusion*; observation* |

scientific disagreement by using a large set of citances across all fields and by developing a set of cues validated by expert assessment.

Other studies of disagreement have been performed in the context of classification schemes of citation function. In an early attempt to categorize types of citations, disagreement was captured as "juxtapositional" and "negational" citations (*Moravcsik and Murugesan, 1975*). However, this scheme was manually developed using a limited sample of papers and citations, and so the robustness and validity of the categories cannot be easily assessed. More recently, scholars have used larger datasets and machine learning techniques to scale citation classifications, often including categories of citations similar or inclusive of disagreement. For example, Teufel, Siddharthan and Tidhar developed a four-category scheme in which disagreement might be captured under their "weakness" or "contrast" citation types (*Teufel et al., 2006*). Bertin and colleagues used n-grams to study location of negative and positive citations, and showed that that the word "*disagree\**" was much less likely to occur than the word "*agree\**", irrespective of papers' sections (*Bertin et al., 2016*). In another study that aimed to identify meaningful citations, Valenzuela, Ha and Etzioni captured disagreement under the "comparison" citation type (*Valenzuela et al., 2015*). Others have sought more coarse categories: Catalini, Lacetera and Oettl classified over 750,000 references made by papers published in the *Journal of Immunology* as either positive or negative, finding that negative references comprised about 2% of all references made (*Catalini et al., 2015*). However, while these machine learning approaches are useful for analyzing large text data, they are also black boxes which can obfuscate issues and limit interpretation of their results.

Building on these studies, we propose a novel approach for the study of disagreement based on a set of manually-validated cue-phrases. We conduct one of the first empirical investigations into the specific notion of *disagreement* in science, and our inclusive definition allows us to capture explicit disagreement between specific

papers as well as traces of disagreement within a field. Our cue-phrase based approach is more transparent and reproducible than black-box machine learning methodologies commonly employed in citation classification, and also extensively validated using over 3,000 citation sentences representing a range of fields. We extend the scale of past analyses, identifying instances of disagreement across more than four million scientific publications.

## Materials and methods

### Data

We sourced data from an Elsevier ScienceDirect corpus that was also used in a previous study (*Boyack et al., 2018*) and that is hosted at the Centre for Science and Technology Studies (CWTS) at Leiden University. This corpus contains the full-text information of nearly five million English-language research articles, short communications, and review articles published in Elsevier journals between 1980 and 2016. The corpus comprises articles from nearly 3,000 Elsevier journals. Given that Elsevier is the largest publisher in the world, this corpus is one of the largest multidisciplinary sources of full-text scientific articles currently available, with coverage of both natural sciences, medical sciences, as well as the social sciences and humanities.

We focus our analysis on sentences containing in-text citations (citances). These citances were extracted from the full-text of articles following the procedure outlined in previous work (*Boyack et al., 2018*). The Elsevier ScienceDirect corpus that was used was constructed in the following way. First, the Crossref REST API was used to identify all articles published by Elsevier. The full-text of these articles was subsequently downloaded from the Elsevier ScienceDirect API (Article Retrieval API) in XML format. Each XML full-text record was parsed to identify major sections and paragraphs (using XML tags), and sentences (using a sentence-splitting algorithm). In-text citations in the main text were identified by parsing the main text (excluding those in

**Table 3.** Being cited in the context of disagreement has little impact on citations in the year following.

For each field, shown are the number of cited papers, as well as for t + 1, t + 2 and t + 3 with t being the year in which a cited paper first featured in the context of disagreement, its average number of received citations, expected number of received citations, and d the ratio between these two values. When d is greater than one, papers cited in the context of disagreement receive more citations in the following year than expected. When d is less than one, they receive fewer citations than expected.

| Scientific field | Number of records | Avg. citations, t + 1 following disagreement | Expected citations, t + 1 following disagreement | $d_{t+1}$ | Avg. citations, t + 2 | Expected citations, t + 2 | $d_{t+2}$ | Avg. citations, t + 3 | Expected citations, t + 3 | $d_{t+3}$ |
|---|---|---|---|---|---|---|---|---|---|---|
| All | 109,545 | 3.03 | 3.08 | 0.983 | 3.02 | 3.05 | 0.990 | 2.96 | 2.98 | 0.993 |
| Bio & Health | 60,707 | 2.73 | 2.81 | 0.969 | 2.68 | 2.75 | 0.974 | 2.56 | 2.65 | 0.966 |
| Life & Earth | 20,581 | 3.43 | 3.35 | 1.023 | 3.55 | 3.42 | 1.038 | 3.63 | 3.44 | 1.056 |
| Math & Comp | 770 | 3.36 | 3.34 | 1.005 | 3.54 | 3.28 | 1.080 | 3.29 | 2.97 | 1.109 |
| Phys & Engr | 18,011 | 3.55 | 3.52 | 1.006 | 3.48 | 3.44 | 1.010 | 3.43 | 3.34 | 1.027 |
| Soc & Hum | 9,476 | 3.04 | 3.11 | 0.979 | 3.20 | 3.28 | 0.975 | 3.30 | 3.40 | 0.971 |

footnotes and figure and table captions). XML records without in-text citations were discarded, and publications from before 1998 were omitted from analysis due to poor availability of full-text records before that year. The resulting dataset consisted of 4,776,340 publications containing a total of 145,351,937 citances, ranging from 1998–2016.

To facilitate analysis at the level of scientific fields, articles in Elsevier ScienceDirect and references cited in these articles were matched with records in the Web of Science database based on their DOI (where available) and a combination of publication year, volume number, and first page number. (The Web of Science database used by CWTS includes the Science Citation Index Expanded, the Social Sciences Citation Index, and the Arts & Humanities Citation Index; other Web of Science citation indices are not included). We used an existing classification of research articles and review articles in the Web of Science created at CWTS. In this hierarchical classification, each article published between 2000 and 2015 and indexed in the Web of Science was algorithmically assigned to a single micro-level scientific field, each of which are in turn members of one of 817 meso-level fields. It is at this meso-level that we perform our most detailed analyses, the categories being fine-grained enough to provide insights into local communities while also large enough to contain a sufficient number of citances. A further benefit of this approach to clustering is that each meso-level field, and each individual publication, can be directly grouped into one of five broad fields:

biomedical and health sciences; life and earth sciences; mathematics and computer science; physical sciences and engineering; and social sciences and humanities. Linking our dataset to this classification system resulted in a subset of 3,883,563 papers containing 118,012,368 citances, spanning 2000–2015. The classification was created algorithmically based on direct citation links between articles, using the methodology introduced by *van Eck and Waltman, 2010* and *Traag et al., 2019*. A visualization of the meso-level classification was created using the VOSviewer software (*van Eck and Waltman, 2010*).

### Operationalizing disagreement

Researchers can disagree for many reasons, sometimes over data and methodologies, but more often because of differences in interpretation (*Dieckmann and Johnson, 2019*). Some of these disagreements are explicitly hostile and adversarial, whereas others are more subtle, such as contrasting findings with past results and theories. We introduce an inclusive definition of disagreement that captures explicit textual instances of disagreement, controversy, dissonance, or lack of consensus between scientific publications, including cases where citing authors are not taking an explicit stance themselves. Our definition distinguishes between two kinds of disagreement, which together capture the diversity of obvious and subtle disagreement in the scientific literature: *paper-level disagreement* and *community-level disagreement*.

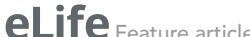

**Figure 3.** Heterogeneity in disagreement across meso-fields. Fine-grained view across 817 meso-level fields, each a cluster of publications grouped and positioned based on their citation links derived from the Web of Science database (see Materials and methods), 2000–2015. The area of each point is proportional to the number of disagreement citances in that field. Overlapping points are an artifact of their position and size, and bear no additional meaning. Color maps to the log ratio of the share of disagreement citances given the mean share across all fields, truncated at 4 x greater and 4 x lower than the mean. Soc & Hum tends to have a greater proportion of disagreement citances, and Math & Comp the least. Other panels show the same data, but highlight the meso-fields in each high-level field. Meso-fields of interest are highlighted, and labels show a selection of journals in which papers in each field are published. Journals listed in labels are representative of each meso-field in the Web of Science, and is not limited to those represented in the Elsevier ScienceDirect data. An interactive version of this visualization is available online at https://tinyurl.com/disagreement-meso-fields.

The online version of this article includes the following figure supplement(s) for figure 3:

**Figure supplement 1.** On average, older papers are less likely to receive a disagreement citation, though this trend does not hold for the "hard" sciences.

**Figure supplement 2.** Distribution of citances by their position in the text of the manuscript, and by field.

**Figure supplement 3.** Little difference in disagreement between men and women.

**Figure supplement 4.** Authors disagree less when citing their own work.

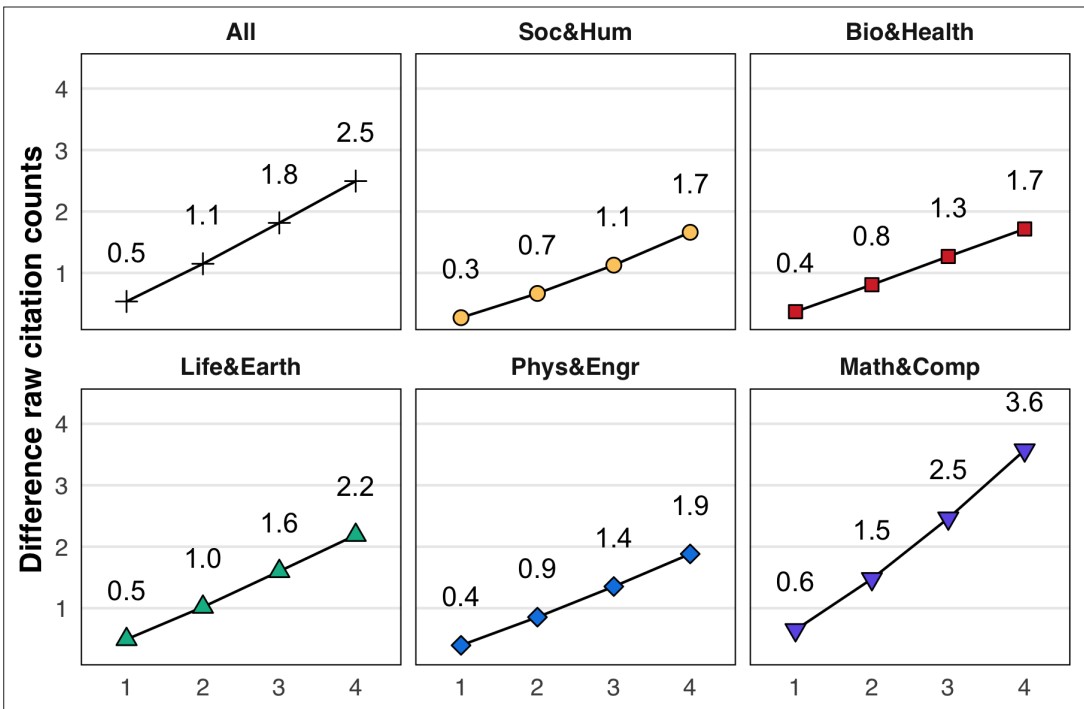

**Figure 4.** Full research articles with a disagreement citance are cited more. The y-axis shows the difference in average citation counts for papers containing at least one disagreement citance, and for papers without. Positive values indicate that publications with disagreement received more citations than those without. Values are shown for the population of publications in each year following publication (x-axis). Shown here for only articles labeled in the Web of Science database as full research articles.

The first, *paper-level disagreement*, occurs when one publication offers a finding or perspective that is (at least partly) incompatible with the perspective of another (even though there may be no explicit contradiction). Consider the following artificial example of a citation sentence explicitly disagreeing with the conclusion of a past study:

> We find that coffee does not cause cancer, contrary to the finding of <ref> that coffee does cause cancer.

*Paper-level disagreement* can also be more subtle. For example, in the following two disagreement sentences, although they do not resolutely contradict one another, the citing and cited publications use models that are based on incompatible assumptions (first sentence), or observe different effects from different data (second sentence):

> Assuming that coffee increases the probability of cancer by 50%, the predicted life expectancy for the Dutch population is 80 years, in contrast to the 85 years proposed by models that assumed coffee does not increase the risk of cancer <ref>.

> Contrary to previous studies that did not observe evidence to support the hypothesis that coffee causes cancer <ref>, our data suggests that drinking coffee increases the probability of cancer by 50%.

*Community-level disagreement*, in contrast, refers to the situation in which a citing publication, without explicitly disagreeing with a cited publication, instead draws attention to a controversy or lack of consensus in the larger body of literature. Including community-level disagreement allows us to identify indirect traces of disagreement in a field, even in the absence of explicit disagreement between the referenced authors, or between the citing and cited papers. Consider the following examples of community-level disagreement; the first notes the disagreement between the referenced studies, and the second cites a single review article indicating disagreement within the field:

> There remains controversy in the scientific literature over whether or not coffee is associated with an increased risk of cancer <refs>.
> A recent review of studies assessing the potential link between coffee consumption

and cancer risk has observed continued controversy <ref>.

Here, we do not differentiate between paper-level or community-level disagreement, including both under our operationalization of disagreement.

### Signal and filter terms

We compose cue-phrases of *signal* terms and *filter* terms. A variety of approaches can be used to generate these terms, and our approach is not dependent on any particular strategy. Here, we create a preliminary set of signal terms through an intensive iterative process of manually identifying, classifying, validating, and deliberating on strategies for retrieving instances of disagreement. This took place over several meetings, utilizing multiple approaches to generate signal words, including sourcing cues used in related work (e.g., *Bertin et al., 2016*; *Chen et al., 2018*), expanding this list with synonyms from online thesauruses, and ranking them by their frequency among citation sentences. This inductive process included several rounds of deliberation, manual annotation, and tests of inter-rater reliability in order to generate a robust list of candidate signal terms. The terms are intended to have high validity, but are not considered comprehensive.

We queried the database for citances containing each of these signal terms (case insensitive), using wildcards to provide for possible variants of terms (e.g., "challenge", "challenged", and "challenges"), excluding generic negation phrases ("no", "not", "cannot", "nor" and "neither" to exclude phrases such as "no conflict"), and for some signal terms excluding citances containing words associated with disciplinary jargon or methods, such as for the signal term "disagreement", which often appears with Likert-scale descriptions (e.g., "scale", "agreement", or "kappa") for survey-heavy fields. The modifications for the signal terms were derived after several rounds of review and validation. In total, citances returned by signal phrase queries comprise 3.10% of the database (n=145,351,937), though their relative occurrence varied dramatically, with the most coming from the "*differ\**" signal term, and the least from "*disprove\**" (see *Table 1*).

In order to more precisely capture valid instances of disagreement and to understand their function within the literature, we also queried for citances containing both the signal terms along with at least one of four sets of *filter* terms, with no more than four words separating signal and filter in the same sentence. As with signal terms,

filter terms were derived from iterative manual efforts of the authors to identify terms most associated with valid instances of disagreement. Four distinct sets of terms were identified, corresponding to explicit mentions of terms relating to past studies, ideas, methods, and results (see *Table 2*). As with signal phrases alone, the relative incidence of signal and filter phrase combinations varies widely (see Table S1 in *Supplementary file 1*). Queries were constructed for each combination of signal term (13 total) and filter term (four total sets), producing 52 combined queries, alongside 13 queries consisting only of standalone signal terms unrestricted by filter terms, for a total of 65 queries.

### Query validation

From each set of results returned by the 65 queries, we selected 50 sentences for validation using simple random sampling without replacement (only 40 citances existed for "no consensus" +"ideas"), resulting in over 3,000 queried sentences. For each query, two coders were randomly selected from among the seven authors on this paper to manually annotate each citance as a valid or invalid instance of disagreement. The label was chosen based only on the text in the citation sentence, without knowledge on the citing paper's title, authors, field of study, or the surrounding text.

Consider the following four example sentences listed below (where (…) indicates the position of cited references and […] indicates additional text not quoted here). The first is invalid because the signal term, "conflict", refers to an object of study, and not a scientific dispute; the second sentence is also invalid because the term "conflicting" refers to results within a single study, not between studies; the third sentence is invalid because "challenge" appears while quoting the cited study; the fourth and fifth sentence are both examples of sentences that would be marked as valid. Similar patterns can be observed for other signal terms, such as *challenge\** (see Table S2 in *Supplementary file 1*).

1. Invalid: "To facilitate conflict management and analysis in Mcr (…), the Graph Model for Conflict Resolution (GMCR) (…) was used."
2. Invalid: "The 4 year extension study provided ambiguous […] and conflicting post hoc […] results."
3. Invalid: "Past studies (...) review the theoretical literature and concludes that future empirical research should 'challenge the assumptions and analysis of the theory'."

4. Valid: "These observations are rather in <u>contradiction</u> with Smith et al.'s […]."
5. Valid: "Although there is substantial evidence supporting this idea, there are also recent <u>conflicting</u> reports (…)."

We assessed the labels for each signal/filter term combination with two measures: percent agreement (% agree) and percent valid (% valid). Percent agreement is the proportion of annotated citances in which both coders agreed on the same label of valid or invalid; this measure provides a simple measure of coder's consensus. Here, percent agreement is justified over more complicated measures (such as Cohen's kappa) due to the small sample of data per signal/filter term combination, and given that there are only two categories and coders.

Most signal/filter term combinations had high agreement (*Figure 1a*). The overall percentage agreement between coders was high, at 85.5%. Given the difficulty of interpreting academic texts, this high percentage agreement demonstrates the robustness of our operationalization of disagreement. The signal term with the highest average agreement was *no consensus* (95.8%). There were only a few combinations with very low percentage agreement, mostly regarding the signal term *questionable,* which had an average lowest average percent agreement (64%); the nature of sentences returned from the *questionable* keyword tended to constitute marginal cases of disagreement. There was virtually no variance between the average percent agreement aggregated across filter terms. However, certain combinations of signal and filter terms were notable in resulting in higher or lower performance. For example, the difference between the highest agreement, *differ* _standalone_ (100%),* and *differ* + methods* (74%) is 26 pereentage points—the addition of filter terms can dramatically impact the kinds of citances returned by the query.

We calculate the percent valid as the percentage of citances annotated as valid by both coders; this provides an intuitive measure of the validity and reliability of a query. Signal/filter term combinations that best capture disagreement should have both high percent agreement and high percent validity. Not all signal/filter term combinations were found to be sufficiently valid (*Figure 1b*). Overall, 61.6% of all coded citances were labeled as valid, with large variance between the most (100%), and the least valid (0%) combinations. The signal term with the highest average validity regardless of filter term was *no consensus*

(94.9%), followed by *controvers** (88.8%) and *debat** (82.4%). Unlike with percent agreement, average validity differs drastically between filter terms, with all having higher average validity than _*standalone*_. The combinations with highest validity are *no consensus + studies* (98%), *no consensus + methods* (98%), and *no consensus _standalone_* (94%) For specific signal terms, the presence of a filter term can have a drastic impact of coded validity; for example, the validity of *contrast** + ideas (80%) is four times greater than of *contrast* _standalone_* and *contrast + methods (20%)*.

The queries that best capture instances of disagreement are those with the highest validity. We choose a validity threshold of 80% and exclude queries with lower validity from subsequent analysis. We also consider several adjustments to the threshold to assess the robustness of our empirical findings. 23 queries sit above this the 80% threshold (*Figure 1c*), including all five *no consensus* and *controvers** queries, four *debat** queries, two *disagree** and *contradict** queries, and one query each for *contrary*, contrast*, conflict*, disprove*,* and *questionable*. Because we prioritized precision, these 23 queries comprise only a fraction of total citances: 455,625, representing 0.31% of all citances in our dataset. We note that citances returned by queries are not exclusive; for example, a citance containing both *controvers** and *no consensus** would count towards both signal phrases. Similarly, a citance returned with the query *controvers* + methods* would also be returned by the *controvers**. Naturally, more general queries, such as *differ** and *contrast** returned a much greater number of citances. Among queries above the 80% threshold, *controvers** and *debat** produce the highest number of citances (154,608 and 150,617 respectively, *Figure 1d*). The 455,625 citances returned by our queries as well as relevant publication and query details are available in Zenodo (*Lamers and Van Eck, 2021*).

## Results

Instances of disagreement, operationalized using the 23 validated queries, accounted for approximately 0.31% of all citation sentences (citances) extracted from indexed papers published between 2000 and 2015 (*Figure 2a*). Disagreement was highest in the social sciences and humanities (Soc & Hum; 0.61%), followed by biomedical and health sciences (Bio & Health; 0.41%), life and earth sciences (Life & Earth; 0.29%); physical sciences and engineering (Phys

& Engr; 0.15%), and mathematics and computer science (Math & Comp; 0.06%).

Our measure shows that disagreement has been relatively constant over time (*Figure 2b*), decreasing at an average rate of about 0.0005 percentage points per year. This is driven by falling disagreement in Phys & Engr (–0.0045 points per year), Soc & Hum (–0.0033 points per year), and Math & Comp (–0.0019 points per year). Phys & Engr stands out not only for its stable decrease each year, but also for its relative size; given a starting rate of one disagreement signal per 529 citances in 2000, by 2015 the rate of disagreement in Phys & Engr fell to one disagreement per 809 citances, a 35% decrease, compared to a 24% decrease for Math & Comp and only a 5% decrease in Soc & Hum. In contrast, disagreement has tended to increase somewhat in Bio & Health ( + 0.0017 points per year) and Life & Earth ( + 0.0018 points per year). These trends are likely not the result of uses of individual queries; for example, *disagree** queries are over-represented in Phys & Engr (see VI in *Figure 2— figure supplement 1*), yet the incidence of these terms is falling or remaining stable (*Figure 2— figure supplement 1*). Similarly, *debat** was over-represented in Soc & Hum and has increased in usage despite slight falling disagreement in the field. That these changes are not confined to any single query suggests that field-level differences represent changes in the level of disagreement within a field rather than linguistic or method-ological artifacts.

### Heterogeneity in disagreement across scientific fields

The more fine-grained meso-fields reveal hetero-geneity within the larger fields (*Figure 3*). Overall, meso-field disagreement followed the same pattern as *Figure 2*, with higher scores in Soc & Hum and lower in Math & Comp. However, some meso-fields stand out. For example, some of the highest rates of disagreement found in Bio & Health meso-fields were in more socially-oriented journals such as Quality of Life Research, Value in Health, and Pharmacoeconomics. Simi-larly, in Math & Comp, the meso-field with the most disagreement contained journals relating to transportation science, a technical field which draws on management studies and other social science literature. This pattern held in Life & Earth, in which a meso-field with a relatively high share of disagreement contained papers in journals such as *Marine Policy*, *Ecology & Society*, and *Forest Policy & Economics*. The high

disagreement in these meso-fields lends support to the hypothesis that regardless of the high-level field, more socially-oriented topics generate a higher level of disagreement. Also notable is that, in Life & Earth, several large fields with relatively high disagreement study the distant geological past or other inaccessible objects of studies, comprised of papers in journals such as the *Journal of Vertebrate Paleontology*, *Creta-ceous Research*, and *Sedimentary Research*. A similar observation can be made in Phys & Engr, where astronomy-related fields featuring jour-nals such as *Planetary and Space Science* and *Theoretical Biology* exhibit above-average rates of disagreement, along with fields pertaining to research into superconductivity. Field-level results must be interpreted cautiously, however, as our signal terms may misclassify citances based on disciplinary keywords and jargon (see Table S3 in *Supplementary file 1*).

### Disagreement by contextual factors

We also investigated the extent to which other factors, including paper age, citance posi-tion, author demographics, and self-citation, relate to a paper's being cited in the context of disagreement.

#### Paper age

Authors may disagree with more recent papers at different rates than older ones. We quan-tify disagreement based on the age of a cited paper papers (relative to the citing paper) and find that, on average, younger papers are more likely to feature in a disagreement citance than older ones, which may indicate that the role of cited literature varies based on its age (*He and Chen, 2018*). Following a brief *bump,* or increase in disagreement (at 05–09 years), older papers tend to be receive fewer disagreement citances (*Figure 3—figure supplement 1*), a pattern driven by field differences. Low consensus, high complexity fields such as Soc & Hum and Bio & Health both exhibit a clear decreasing pattern, with falling disagreement as the paper ages. Life & Earth, in the middle of the hierarchy, repeats this pattern, but only after a period of stability in disagreement in the first ten years. Disagree-ment instead steadily increases over time in high consensus and low complexity fields such as Phys & Engr and Math & Comp.

#### Position in the paper

Disagreement is not equally likely to occur throughout a paper. Investigating the distribution

of disagreement citances across papers, we find that they are far more likely to occur in the beginning of a paper, likely in the introduction, and then towards the end, likely the discussion section (*Figure 3—figure supplement 2*), corresponding to previous observations of disagreement cue phrases in PLoS journals (Bertin et a., 2016), and likely indicating a unique argumentative role of disagreement. The precise patterns differ by field. For example, in Soc & Hum, disagreement citances are more evenly distributed through the first 40% of the paper, whereas in Bio & Health and Life & Earth disagreement citations are more likely to appear near the end of a paper. While these field level differences may reflect differences in how fields use citations, they are more likely the result of distinct article formats across fields (i.e., long literature reviews in Soc & Hum).

## Gender of Citing-Paper Author
Men and women authors may issue disagreement citances at different rates. To investigate this, we infer a gender for the first and last authors of papers issuing a disagreement citance and published after 2008, determined based on the author's first name as in past work (*Larivière et al., 2013*). Overall, there is little difference in the rate at which disagreement is introduced by men and women first and last authors (*Figure 3—figure supplement 3*). The one exception is Math & Comp, in which women last authors issue 1.2 times more disagreement citations than men, though the rate of disagreement is small, and driven by a small number of instances.

## Self-citation
As a confirmation of overall validity, we measure the rate of disagreement by instances of self-citation and non-self-citation. We expect that authors will be less likely to cite their own work within the context of disagreement. Indeed, we find that the rate of disagreement for non-self-citations is 2.4 times greater than for self-citations (*Figure 3—figure supplement 4*), demonstrating that our indicator of disagreement affirms expectations. The field with the largest difference is Bio & Health (2.5 times greater), followed by Phys & Engr (2.2 times greater), Math & Comp (2.2 times greater), Life & Earth (1.9 times greater), and finally, Soc & Hum (1.6 times greater).

## *Robustness*
To test the robustness of our results, we compare findings using the 23 queries with greater than 80% validity to those using the 36 queries with

greater than 70% validity. The new queries include *contradict\* _standalone_, contrary + studies, contrary + methods, conflict\* + results, disagree\* + methods, disagree\* + ideas, disprov\* + methods, disprov\* + ideas, refut\* + studies, refut\* + results, refut\* + ideas, debat\* + ideas,* and *questionable + ideas.* Queries above the 80% validity cutoff account for 455,625 citances; the addition of 13 queries above the 70% cutoff bring this total to 574,020.

We find that our findings are robust whether using an 80% or 70% validity cutoff. Relaxing the validity cutoff results in including more citances, inflating the share of disagreement across all results. However, the qualitative interpretation of these results does not change (Table S4). The 80% and 70% cutoffs both produce the same ordering of fields from most to least disagreement. Similarly, the ordering of fields from high-to-low disagreement holds between the 80% and 70% cutoff for all quantities presented here, including the average change per year, the ratio of disagreement between non-self-citation and self-citation, and the average change in disagreement per age bin. Some fields gain more from these new queries than others, manifesting in more or less intense field differences. For example, Soc & Hum gains a full 17 percentage points in overall disagreement with the 70% threshold, with the increase across all fields at only eight points. Similarly, the ratio of non-self-citation to self-citation is 2.2 x for Math & Comp with the 80% cutoff, but only 1.3 x for the 70% cutoff. Future work may find that different thresholds are more appropriate across fields, depending on their distinct patterns of discourse.

## *Disciplinary differences in query results*
It is worthile to consider how specific queries manifest across fields, which can give insights into their unique uses of language and disagreement. We consider the incidence of each query compared to an expected value, given the distribution of citances across all high-level fields. This is necessary as the number of publications varies across fields. For example there are far more publications from Bio & Health than in other fields, accounting for a total of 47.5% of all publications indexed in the Web of Science Database; in contrast, publications in Math & Comp comprise a far smaller proportion of the database, accounting for only 3.1%.

Even accounting for the different number of publications per field, we still observe that some signal terms appear more in certain fields than

expected, often as a result of differences in disciplinary jargon, topics, and norms (*Figure 1— figure supplement 1b*). For example, there are more *conflict\** citances than expected in Soc & Hum, where it often appears in relation to conflict as a topic of study, such as the study of international conflict, conflict theory, or other interpersonal conflicts (line I in Table S3). Similarly, *disprov\** citances appear more often in Math & Comp, where disprove is often used in relation to proving or disproving theorems and other mathematical proofs (line II in Table S3). Other notable differences are *controvers\** citances appearing more often in Bio & Health, *debat\** appearing most often in Life & Earth, and *disagree\** appearing most in Phys & Engr.

Filter terms are also not randomly distributed across fields (*Figure 1—figure supplement 1c*). For example, the +*ideas* filter term appears more often than expected in Soc & Hum, possibly as a result of disciplinary norms around use and discussion of abstract theories and concepts (line III in Table S3). In contrast, + *methods* is over-represented in Phys & Engr and Math & Comp, likely a result of these field's focus on methods and technique (line IV in Table S3). Notably, + *studies* and + *results* are under-represented among Math & Comp publications, whereas + *ideas* and + *methods* are underrepresented among papers in the Bio & Health.

The complexity of disciplinary differences between queries is made apparent when examining combinations of signal and filter phrases (*Figure 1—figure supplement 1d*). While there are no obvious or consistent patterns across fields, there are notable cases. For example, compared to all other fields, *controvers\** citances are over-represented in Bio & Health (line V in Table S3), except for *controvers + ideas,* which is instead slightly over-represented in Life & Earth. In contrast, *disagree\** citances are under-represented in Bio & Health, but over-represented in Life & Earth and Phys & Engr (line VI in Table S3). In some cases, the specific signal + filter term combination has a massive effect, such as *no consensus + ideas*, which is heavily over-represented in Soc & Hum (line VII in Table S3), whereas all other signal and filter term combinations are under-represented. Similarly, *contradict\* + ideas* and *contradict\* + methods* are over-represented in Math & Comp (line VIII in Table S3), whereas + *results* and + *studies* are underrepresented. Similar intricacies can be found across the 325 combinations of cue-phrase and field, demonstrating the importance that

field plays in the utility and significance of our signal and filter terms.

Especially at the fine-grained field level, methodological artifacts can drive differences we observe between meso-fields. For example, in Soc & Hum, one of the meso-fields with the most disagreement was composed of papers from journals such as *Political Studies* and *International Relations*—journals and fields for which "debates" and "conflicts" are objects of study, which could confound the *debat\** and *conflict\** signal terms. This is demonstrated by the following invalid citation sentences:

1. "Since the late-1990s, there has been even less room for **debate** within the party (...)."
2. "Indeed, this whole idea harkens back to the badges of slavery of the 13th Amendment and the **debate** in (...)."
3. "In political behaviour literature, we refer to such **conflictive** opinions as "ambivalence" (...)."
4. "In politics as usual, people often do not like to see the **conflicts** and disagreements common to partisan debate (...)."

Even though terms such as "public debate", and "parliamentary debate" were excluded (*Table 1*), the *debat\** signal terms were over-represented in Soc & Hum (*Figure 1—figure supplement 1b*); *conflict\** was also overrepresented to a lesser extent. Interpretation of the results for main and meso-fields needs to be moderated by these, and other confounding artifacts.

## Assessment of individual papers

We perform a qualitative investigation of the individual papers that issued the most disagreement citations, and which were cited most often in the context of disagreement. First, we examine the citing paper perspective, that is those papers that issued the most citances (Table S5 in *Supplementary file 2*). These top papers demonstrate how methodological artifacts can contribute to these more extreme examples. For example, one of these papers considers the pedagogical and evaluative potential of debates in the classroom (*Doody and Condon, 2012*); the *"debat\*"* signal term incorrectly classifies several citations as evidence of scientific disagreement. However, other papers offer interesting instances of disagreement, and exemplify lessons that should be considered in its study. For instance, one such paper concerns meteorite impact structures (*French and Koeberl, 2010*) and includes discussion on the controversies in the field. Another is a review article arguing for multi-target agents

for treating depressive states (**Millan, 2006**), and catalogs the controversies around the topic. Yet another is a book on *Neurotoxicology and Teratology*, misclassified as a research article in the database, which illustrates how the length of an article can contribute to its likelihood of issuing a disagreement citation (**Kalter, 2003**).

Considering the cited paper perspective—those papers that received the most paper-level disagreement citations or were referenced the most in the context of community disagreement—reveals clear instances of disagreement in the literature. Many of the studies receiving the most disagreement citances (Table S6 in **Supplementary file 2**) relate to a single longstanding scientific controversy in the earth sciences concerning the formation of the North China Craton, a tectonic structure spanning Northern China, Inner Mongolia, the Yellow Sea, and North Korea. This list of most-disagreed-with papers also includes a literature review that is cited as an exemplar of controversy, here regarding the existence of "lipid rafts" in cells (**Munro, 2003**), and a paper on fMRI research that is heralded as a methodological improvement in the field, and is often cited to draw a contrast with other methods (**Murphy et al., 2009**). A more thorough discussion of papers that issue and receive the most disagreement can be found in **Supplementary file 2**.

### Disagreement and citation impact

We also explore whether disagreement relates to citation impact; whereas previous analysis revealed a positive relationship between conflict and citation (**Radicchi, 2012**), our preliminary results do not find evidence of increased citation, at least in the years immediately following the disagreement (**Table 3**). We arrived at this observation by comparing the number of citations received in year $t + 1$ for papers that featured in a disagreement citance for the first time in year $t$, with the average number of citations received in year $t + 1$ by papers that received the exact same number of citations in year $t$. This over- or under-citation of individual papers that encountered disagreement can then be aggregated to arrive at the average over- or under-citation following disagreement.

We define $t$ as the time in years since publication and $c$ as the number of citations a paper received at time $t$. We calculate for each combination of $t$ and $c$ the number of papers $p_{c,t}$ that were first cited in the context of disagreement at time $t$ when they held $c$ citations. Using these,

we calculate the number of citations received by these papers in the year following publication, averaged across all combinations of $t$ and $c$,

$$\bar{c}_{next,disagreement} = \frac{\sum_c \sum_t p_{c,t} * \bar{c}_{next,disagreement,c,t}}{\sum_c \sum_t p_{c,t}}$$

In the same way, we also calculate the expected number of citations, defined using the average number of citations received by papers that received $c$ citations in year $t$, regardless of whether they were cited by a disagreement citation.

$$\bar{c}_{next,expected} = \frac{\sum_c \sum_t p_{c,t} * \bar{c}_{next,expected,c,t}}{\sum_c \sum_t p_{c,t}}$$

We calculated $d$ as the ratio of these two values. When greater than one, it indicates that papers received more citations than expected in the year after having been cited in the context of disagreement. A value less than one indicates that papers with a disagreement citation received fewer citations in the year following.

$$d = \frac{\bar{c}_{next,disagreement}}{\bar{c}_{next,expected}}$$

The results of this analysis (**Table 3**) show that being cited in a context of disagreement has little to no effect on the citations received by papers in the year following their citation (or not) in the context of disagreement. Extending the analysis to citations received in year $t + 2$ and $t + 3$ yielded similar null results.

Papers that themselves contain disagreement citances, however, tend to receive more citations over their lifespan. To demonstrate this, we examine the 3.5% (n = 126,250) of publications that contain at least one disagreement citance in their text. Across all publications, those with at least one disagreement citance tended to receive more citations than those without disagreement in the first four years, beginning with one additional citation in the first year following publication, and expanding to a difference of about 4.7 citations by the fourth year (**Figure 4**), an effect that varies, yet is qualitatively consistent across all fields. This effect may be confounded by article type—for example, review articles are over-represented in terms of disagreement—24.6% of all review articles contain a disagreement citance—and review articles are also known to be more highly cited (**Miranda and Garcia-Carpintero, 2018**). While

excluding review articles does shrink this gap, the citation count for full research articles (85% of all publications) remains 2.5 citations higher for those with a disagreement citance than for those without.

We note that these results are confounded by our umbrella definition of disagreement, which does not differentiate between paper-level and community-level disagreement. Paper-level disagreement—when the author of the citing paper explicitly contrasts their study with another—is a straightforward example of issuing (by the citing paper) and receiving (by the cited paper) disagreement. Community-level disagreement, in contrast, either involves a citing author rhetorically positioning two or more papers as being in disagreement, or citing a single past paper such as a review, as evidence of the controversy surrounding a topic. While these two cases offer evidence of disagreement in the field, their potential for identifying specific, controversial papers is less clear (*Radicchi, 2012*). Future research should aim to disentangle paper-level and community-level disagreement, and understand their varying relationship to citation impact.

## Discussion

When it comes to defining scientific disagreement, scholars disagree. Rather than staking out a specific definition, we adopt a broad operationalization of disagreement that incorporates elements of Kuhn's accumulation of anomalies and paradigm shifts (*Kuhn, 1996*), Latour's controversies (*Latour, 1988*), and more recent notions of uncertainty (*Chen et al., 2018*) and negative citations (*Catalini et al., 2015*). By bridging these past theories, we quantify the rate of disagreement across science. Roughly 0.31% of all citances in our dataset are instances of disagreement, a share that has remained relatively stable over time. However, this number is much smaller than in past studies—such as the 2.4% for so-called "negative" references (*Catalini et al., 2015*), and the estimated 0.8% for "disputing" citations (*Nicholson et al., 2021*). This is explained by our operationalization of disagreement, which although conceptually broader than negative or disputing citations, is narrowed to only 23 queries to prioritize precision. Moreover, studies differ in corpus used, most often covering only one journal or field, compared to our large multidisciplinary corpus. The strength of our analysis is not the absolute incidence of disagreement, but its relative differences across disciplinary and social contexts.

Disagreement across fields can be interpreted using several theoretical frameworks. Differences in disagreement might stem from the *epistemic* characteristics of fields and their topics of study. For example, Auguste Comte proposed that fields are organized based on the inherent complexity of their subject matter (*Comte, 1856*). We reinforce this "hierarchy of sciences" model, finding that disagreement is highest in fields at the top of the hierarchy, such as the social sciences and humanities, and lowest in fields at the bottom of the hierarchy, such as physics and mathematics.

While the hierarchy of sciences model is well-grounded theoretically (*Cole, 1983*) and bibliometrically (*Fanelli, 2010*; *Fanelli and Glänzel, 2013*), other frameworks may be equally useful in understanding disagreement across fields. For example, the structural characteristics of fields may explain their differences in disagreement. One such characteristic is how reliant the field is on Kuhnian paradigms (*Kuhn, 1996*); so-called "hard" sciences, such as physics, may have strong theoretical paradigms and greater consensus (less disagreement) than "soft" sciences such as those in the social sciences and humanities (*Biglan, 1973*).

Changes in these structural characteristics may also contribute to the temporal evolution of disagreement. For instance, the decrease of disagreement in physics and engineering may be due to a transition into a period of "normal" science (*Kuhn, 1996*), as it has been previously argued for certain sub-fields (*Smolin, 2007*). Increase in collaboration (*Wuchty et al., 2007*) may also affect the trends, as consensus has to be reached among a larger body of individuals during the research process.

Social sciences and humanities have other characteristics that might be associated with more common or more intense conflicts, including low centralization of resources and control over research agendas, high diversity in their audiences and stakeholders, and limited standardization of methods and theories (*Whitley, 2000*). A field's *cultural* characteristics also play a role in its norms of disagreement. Fields have different norms when it comes to consensus formation and the settling of disputes (*de Cetina and Reyes, 1999*), and some fields even value disagreement as an important element of scholarship. For instance, a cultural norm of "agonism", or ritualized adversarialism, is common in many humanities fields, wherein one's arguments are framed in direct opposition to past arguments (*Tannen, 2002*).

Fields also have distinct cultures of evaluation, which shapes how they judge each other's work and impacts whether they are likely to reach consensus (*Lamont, 2009*). Of course, epistemic, structural, and cultural characteristics of fields are all inter-related—cultural practices emerge in part from structural characteristics of a field, such as access to expensive instruments, which in turn are related to the epistemic aspects of the object of study. Our data does not allow us to disentangle these relationships or argue which is most appropriate, but each offers a useful lens for understanding why disagreement might differ between fields.

Expanding our analysis into a more fine-grained classification of fields reveals greater detail into where disagreement happens in science. We observed that socially-oriented meso-level fields tended to have a higher rate of disagreement, no matter their main field. For example, meso-fields concerning healthcare policy had higher rates of disagreement than others in the biomedical and health sciences whereas the meso-field concerning transportation science had a higher rate of disagreement than all others mathematics and computer science. Though these fields draw on the expertise of the "hard" sciences, they do so in order to study social processes and address social questions.

In the life and earth sciences, disagreement was especially high in meso-fields that study the earth's geological and paleontological history. In these fields, much like in the social sciences, researchers cannot easily design experiments, and so progress instead comes from debate over competing theories using limited evidence and reconstructed historical records. This is exemplified by paleontology, in which a 2017 paper sparked controversy and forced a re-interpretation of the fossil record and a 130-year-old theory of dinosaur evolution (*Baron et al., 2017*; *Langer et al., 2017*). Similarly, our approach identified a major controversy in the earth sciences—the formation of the North China Craton—again illustrating how reliance on historical records might exacerbate disagreement. These cases illustrate the heterogeneity of disagreement in science, and illustrate that existing theoretical frameworks, such as the hierarchy of science, can oversimplify the diversity of cultural norms and epistemic characteristics that manifest at more fine-grained levels of analysis.

Our approach comes with limitations. First, our method captures only a fraction of textual disagreements in science. This is partly due to our prioritization of precision over recall, having

removed cue-phrases with low validity. Our lists of signal and filter terms are also non-exhaustive, and so their extension in future research would identify more instance of disagreement. Given our focus on citances, we are not able to identify traces of disagreements that occur without explicit reference to past literature, or those that can only be classified as disagreement with surrounding sentences as context. Some disagreements may also be too subtle, or rely on technical jargon, such that they cannot be identified with our general signal terms. Moreover, our measure does not capture non-explicit disagreements, or scientific disagreements occurring outside of citances, such as in conferences, books, social media, or in interpersonal interactions. For these reasons, our measure of disagreement may over- or under-represent disagreement in particular fields, and should be considered when evaluating results.

Second, in spite of its overall precision, our approach returns many false positives in particular disciplinary contexts. For example, the signal term *conflict** matches to topics of study and theories in the fields of sociology and international relations (e.g., "ethnic conflicts", "Conflict theory"). In other instances, a signal term can even match an author's name (as in the surname "Debat", as in *Debat et al., 2003*). We also find that these artifacts are over-represented among the papers that issued the most disagreement citances, and those that were most often cited in the context of disagreement (see Appendix 6). However, given our extensive validation, these artifacts remain a small minority of all disagreement sentences identified, though they should be considered when interpreting disagreement in small sub-fields.

Finally, our inclusive definition of disagreement homogenizes disagreement into a single category, whereas there are many kinds of disagreement in science. For example, the ability to differentiate between paper-level and community-level disagreement could lend insight into how conflict and controversy manifest in different fields. This definition could also be developed to differentiate further between types of disagreement: for example, past citation classification schemes have differentiated between "juxtaposition" and "negational" citations (*Moravcsik and Murugesan, 1975*), or between "weakness" and "contrast" citations (*Teufel et al., 2006*).

Despite these limitations, our framework and study have several advantages. First, in contrast to keyword-based analyses, our approach provides

a nuanced view of disagreement in science, revealing the differences in disagreement not only between signal terms, but also based on filter terms. This drives the second advantage of our approach—that its inherent *transparency* allows us to easily identify confounding artifacts such as when a signal term is an object of study (i.e., "international *conflicts*", "public *debate*"), when it relates to disciplinary jargon (i.e., "*disproving* theorems" in Mathematics, or *"strongly disagree"* in survey studies that use Likert scales) or when the keyword is part of a proper name (i.e., "work by *Debatin* et al.,"). These issues are a concern for any automated analysis of scientific texts across disciplines—the usage and meaning of words varies across fields. In contrast to black-box style machine learning approaches, ours is transparent and can easily be validated, interpreted, replicated, and extended.

Finally, by being open and transparent, our approach is easily adjustable to different contexts. Our initial identification of keywords was the result of an iterative process of exploration and validation, which eventually resulted in a non-exhaustive set of signal terms, filter terms, exclusions, and then a final set of validated queries. Any step of this process can be tuned, extended, and improved to facilitate further studies of scientific disagreement—new signals or filters can be introduced, queries can be modified to be even more precise, and the threshold of validity changed; here, for example, we assessed our results by setting a more inclusive threshold for which queries constitute disagreement, and find the results remain robust (see Table S1 in *Supplementary file 1*). To assist in further efforts to validate and extend our work, we have made annotated sentences and code that can reproduce this analysis publicly available at github.com/murrayds/sci-text-disagreement (copy archived at swh:1:rev:b361157a9cfe-b536ca255422280e154855b4e9a3, *Murray, 2021*).

Whereas black-box machine learning approach have many strengths (e.g. *Rife et al., 2021*), ours is transparent and intuitive. Its transparency allows to easily identify terms that have field-specific meanings, which may be obfuscated in black-box approaches. Our approach is also reproducible and can be refined and extended with additional signal and filter terms. The portability of our queries also mean that they can readily be applied to other full-text data. The general method of generating and manually validating signal and filter terms can also be applied to other scientific phenomena, such as detecting uncertainty (*Chen et al., 2018*), negativity (*Catalini et al., 2015*), discovery (*Small et al., 2017*), or an expanded framework of disagreement (*Moravcsik and Murugesan, 1975*).

Future research could refine and extend these existing queries and link them to different conceptual perspectives on disagreement in science. Such work could build on our analyses of the factors that may affect disagreement, including gender, paper age, and citation impact. Disagreement is an essential aspect of knowledge production, and understanding its social, cultural, and epistemic characteristics will reveal fundamental insights into science in the making.

## Acknowledgements
We thank Yong-Yeol Ahn, Staša Milojević, Alessandro Flammini, Filippo Menczer, Dashun Wang, Lili Miao, participants of the "A scientometric analysis of disagreement in science" seminar held at CWTS at Leiden University, and the editor and reviewers for their helpful comments. We are grateful to Elsevier for making the full-text data available to us.

**Wout S Lamers** is in the Centre for Science and Technology Studies, Leiden University, Leiden, Netherlands
w.s.lamers@cwts.leidenuniv.nl
http://orcid.org/0000-0001-7176-9579
**Kevin Boyack** is at SciTech Strategies, Inc, Albuquerque, United States
http://orcid.org/0000-0001-7814-8951
**Vincent Larivière** is at École de bibliothéconomie et des sciences de l'information, Université de Montréal, Montréal, Canada
http://orcid.org/0000-0002-2733-0689
**Cassidy R Sugimoto** is in the School of Public Policy, Georgia Institute of Technology, Atlanta, United States
**Nees Jan van Eck** is in the Centre for Science and Technology Studies, Leiden University, Leiden, Netherlands
http://orcid.org/0000-0001-8448-4521
**Ludo Waltman** is in the Centre for Science and Technology Studies, Leiden University, Leiden, Netherlands
http://orcid.org/0000-0001-8249-1752
**Dakota Murray** is in the School of Informatics, Computing, and Engineering, Indiana University, Bloomington, United States
dakmurra@iu.edu
http://orcid.org/0000-0002-7119-0169

*Author contributions:* Wout S Lamers, Conceptualization, Formal analysis, Investigation, Methodology, Software, Visualization, Writing – original draft, Writing – review and editing; Kevin Boyack, Conceptualization, Investigation,

Methodology, Writing – original draft, Writing – review and editing; Vincent Larivière, Conceptualization, Investigation, Methodology, Writing – review and editing; Cassidy R Sugimoto, Conceptualization, Investigation, Methodology, Writing – review and editing; Nees Jan van Eck, Conceptualization, Data curation, Formal analysis, Investigation, Methodology, Software, Visualization, Writing – review and editing; Ludo Waltman, Investigation, Methodology, Writing – review and editing; Dakota Murray, Conceptualization, Formal analysis, Investigation, Methodology, Project administration, Software, Visualization, Writing – original draft, Writing – review and editing

*Competing interests:* The authors declare that no competing interests exist.

### Funding

| Funder | Grant reference number | Author |
|---|---|---|
| Air Force Office of Scientific Research | FA9550-19-1-039 | Dakota Murray |
| Canada Research Chairs | | Vincent Larivière |

The funders had no role in study design, data collection and interpretation, or the decision to submit the work for publication.

### Decision letter and Author response
Decision letter https://doi.org/10.7554/eLife.72737.sa1
Author response https://doi.org/10.7554/eLife.72737.sa2

## Additional files

### Supplementary files
• Transparent reporting form
• Supplementary file 1. Tables S1 – S4.
• Supplementary file 2. Papers with most disagreement citances and papers most often cited in the context of disagreement.

### Data availability
The Elsevier full-text data and the Web of Science bibliographic data used in this study were obtained from proprietary data sources. We are not allowed to share the raw data on which our study is based. We do have permission from Elsevier to share a data set containing the 455,625 citing sentences identified using our disagreement queries. This data set is available in Zenodo (Lamers & Van Eck, 2021, https://doi.org/10.5281/zenodo.5148058). Source data and visualization code for all figures is available at github.com/murrayds/sci-text-disagreement (copy archived at

swh:1:rev:b361157a9cfeb536ca255422280e154855b4e9a3).

The following dataset was generated:

| Author(s) | Year | Dataset URL | Database and Identifier |
|---|---|---|---|
| Lamers WS, Van Eck NJ | 2021 | https://zenodo.org/record/5148058 | Zenodo, 10.5281/zenodo.5148058 |

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
