## [Decision Letter]

**Decision letter after peer review:**

Thank you for submitting your article "Measuring Disagreement in Science" to *eLife* for consideration as a Feature Article. Your article has been reviewed by three peer reviewers, and the evaluation has been overseen by the *eLife* Features Editor. The following individuals involved in review of your submission have agreed to reveal their identity: Chaomei Chen; Iana Atanassova.

The reviewers and editors have discussed the reviews and we have drafted this decision letter to help you prepare a revised submission.

Summary:

The paper focuses on a study of citation instances associated with disagreements in scientific publications. Qualified citation instances (citances) are identified based on two sets of terms: signal terms and filter terms. Signal terms indicate some forms of disagreement such as controversies and debates, whereas filter terms characterize common elements of scientific publications such as studies, ideas, methods, and results. Citances are validated by two coders from the authors. ScienceDirect articles are matched to the Web of Science articles so that matched articles can be classified into 800+ meso-level fields. The distribution and the rate of appearances of valid citances are presented among other quantitative and qualitative analyses.

The topic is important, and one important strength of the paper is the size of the dataset that is processed (over 4M papers in full text) and also the fact that the papers have been already classified into disciplinary fields and 817 meso-level fields by the previous work of the authors. This constitutes a standpoint that allows to perform large scale analyses and draw conclusions for the different disciplines.

The background of the work is presented generally well, although more detailed comparisons and differentiations are desirable to position the study in the literature in terms of the scope and depth with reference to relevant works. The method and the execution of the study is generally in good order. On the other hand, the presentation, especially some of the interpretations and some of the claims, needs attention in a number of places.

Essential revisions:

1) The title needs attention for a number of reasons. "Measuring Disagreement in Science" is an overstatement because, as the authors state in the Discussion section, their study is rather "an attempt at measuring" or "a proposal for a method of measurement".

Also, "scientific literature" would be more appropriate than "science" because the study is limited to scientific literature and other forms of scientific inquiries are beyond the scope of the study.

Similarly, "disagreement in citation contexts" would be more appropriate than "disagreement" because many other forms of disagreement are outside the scope of the study.

2) More details are needed in some places to allow other researchers to replicate the procedure. For example:

Line 138: Why was the Crossref API used to identify articles when the ScienceDirect database is hosted at CWTS?

Line 152: Why are these meso-level fields used? Are there alternatives?

Line 193: What are the sources of the preliminary signal terms? What are the specific steps for obtaining synonyms, e.g., using any standard and widely available resources such as WordNet?

Line 213. "filter terms within a four-word window of the signal" This expression seems a bit of ambiguous. Does it mean a filter term must be found no more than two words away from the signal term as illustrated by the following scenarios?

Some four-word windows of {signal}:

{filter} word1 word2 {signal}

{signal} word1 word2 {filter}

Presumably such windows do not contain sentence boundaries, e.g., "… word1 {filter}. {signal} word2 …" (since the citances are single sentences, correct?). If that is the case, it should be helpful if you can clarify this explicitly.

Line 222: How exactly was the selection of citances randomized?

3) The authors have cleverly shifted the issue of negative citation to the broader concept of disagreement. They have analyzed a corpus of sentences that all contain a citation, and their aim is to capture both paper-level and community-level disagreement. However, the example of community-level disagreement given Line 188 is not necessarily an instance of disagreement between the cited works/authors, as the authors state it. It might be (it is, actually, most of the times) a list of works that do mention the existence of a debate/controversy about a topic. I think it is a typical example of a citing author agreeing with the cited authors that there is disagreement on a topic ( = agreeing on disagreement). Choosing sentences with citations and cue terms marking disagreement does not in any way guarantee that the authors are expressing disagreement with the works they cite, but merely that there is disagreement in the literature on a topic at a given time. One may therefore ask what is the point of restricting the analysis to sentences with citations and not addressing the problem from the point of view of citation polarity or function, especially if the aim is to group together paper-level and community-level disagreement. I think the citations have the same function here as filter terms, but cannot be seen as the targets of the disagreement, as they are indeed line 373 and in some sections of the Supporting Information part.

4) I understand that precision is the priority in this study. Nevertheless, some equally useful vocabularies are not included in the list in Table 2. For example, commonly used terms that are missing from the list for 'studies' include research, investigation, inquiry, just to name a few. A more extensive list would certainly improve the hit rate. Similarly, some important omissions from the list for 'ideas' include concept and claim.

Please either redo this part of the analysis with more terms, or discuss how this is a limitation of the study.

5) Related to point 4, a quick review of the available dataset and the limitations provided by the authors show that the precision may not be good either. The authors claim that false positives are marginal. They may be right. But when it comes to making analyses at the journal level, this can distort the result somewhat. I will take just one example for lack of space: Figure 3, Soc and Hum, Electoral studies (which is emphasized in the figure) -> with a manual check, I identify 37% of the citations as false positives for this journal. The use of syntactic parsing could have filtered out the false positives from all the examples citing "presidential debates" for example. The problem is that we don't know the amount of false positives at all.

6) Is there an underlying principle that leads to the distinction between the paper- and community-level disagreements?

7) Line 260: Percent valid is defined as the percentage of citances labeled as valid by both coders.

This seems very subjective. If different individuals were chosen as the coders, then we may end up with possibly quite different results because it is quite conceivable that we can find another pair of coders who may make different judgements. Are there alternatives to make this part of the procedure more systematic and reproducible?

8) Line 402-403: prioritize precision … relatively rare.

If the methodology prioritizes precision, then your results cannot be used to support a conclusion that it is relatively rare because one of the many possible consequences of your priority choice is to lower the rate of appearances of qualified instances. If you expand the signal terms and filter terms, then the rate is likely to increase.

9) Line 420: a field is increasingly becoming consensual … (Smolin, 2007).

This interpretation is not particularly convincing. A consensual field is a dying field unless it finds new driving forces. It is essential to maintain the healthy growth of a field as well as to the career and social dynamics of individual researchers who choose the work in the field. On the other hand, consensus at lower levels such as topic levels can be reached without fundamentally damaging the growth of a field. I wonder what you would see if you normalize instances by article and/or by all citances (including non-disagreement citances).

10) Line 474. The framework needs more clarification. For example, how is it related to previous works on negations and uncertainties? For example, manually crafted signal terms vs. black-box approaches, especially for potentially subsequent expansions, coder-based validity vs information theoretic approaches, citer-citee disagreement vs incommensurable paradigms (one may never cite the other from an incommensurable paradigm), and the prevalence of disagreements expressed outside citances.

11) Please revise the manuscript at appropriate places to address the following points:

i) the study was restricted to English-language articles.

ii) the absence of a citation can be an even stronger marker of disagreement than a negative citation.

iii) disagreements in science can take place in other venues, such as conferences, and in other article types (such as book reviews).

---

## [Author Response]

Essential revisions:1) The title needs attention for a number of reasons. "Measuring Disagreement in Science" is an overstatement because, as the authors state in the Discussion section, their study is rather "an attempt at measuring" or "a proposal for a method of measurement".Also, "scientific literature" would be more appropriate than "science" because the study is limited to scientific literature and other forms of scientific inquiries are beyond the scope of the study.Similarly, "disagreement in citation contexts" would be more appropriate than "disagreement" because many other forms of disagreement are outside the scope of the study.

Thank you for this suggestion. We agree that the title of our project needs to be changed to better reflect the nature of our work. However, we also believe that our paper it relevant to the broad and diverse audience at *eLife*, and that an overly-technical title would limit its appeal.

Recognizing this tension, we have updated our title to read “Investigating Disagreement in the Scientific Literature”, which we believe appropriately balances the interests of the reviewers and the potential readers of our work.

2) More details are needed in some places to allow other researchers to replicate the procedure. For example:Line 138: Why was the Crossref API used to identify articles when the ScienceDirect database is hosted at CWTS?Line 152: Why are these meso-level fields used? Are there alternatives?Line 193: What are the sources of the preliminary signal terms? What are the specific steps for obtaining synonyms, e.g., using any standard and widely available resources such as WordNet?Line 213. "filter terms within a four-word window of the signal" This expression seems a bit of ambiguous. Does it mean a filter term must be found no more than two words away from the signal term as illustrated by the following scenarios?Some four-word windows of {signal}:{filter} word1 word2 {signal}{signal} word1 word2 {filter}Presumably such windows do not contain sentence boundaries, e.g., "… word1 {filter}. {signal} word2 …" (since the citances are single sentences, correct?). If that is the case, it should be helpful if you can clarify this explicitly.Line 222: How exactly was the selection of citances randomized?

Thank you for these suggestions. We have made the following edits to our manuscript, which have improved the clarity of our procedure

1. We have updated the text to clarify the nature of our database and the procedure used to originally collect the data and build our database: “We sourced data from an Elsevier ScienceDirect corpus that was also used in a previous study (Boyack et al., 2018) and that is hosted at the Centre for Science and Technology Studies (CWTS) at Leiden University. (…) The Elsevier ScienceDirect corpus that was used was constructed in the following way. First, the Crossref REST API was used to identify all articles published by Elsevier. The full-text of these articles was subsequently downloaded from the Elsevier ScienceDirect API (Article Retrieval API) in XML format.”

2. We have expanded on the selection of the meso-level fields: “In this hierarchical classification, each article published between 2000 and 2015 and indexed in the Web of Science was algorithmically assigned to a single micro-level scientific field, each of which are in turn members of one of 817 meso-level fields. It is at this meso-level that we perform our most detailed analyses, the categories being fine-grained enough to provide insights into local communities while also large enough to contain a sufficient number of citances. A further benefit of this approach to clustering is that each meso-level field, and each individual publication, can be directly grouped into one of five broad fields: *Biomedical and Health Sciences*, *Life and Earth Sciences*, *Mathematics and Computer Science*, *Physical Sciences and Engineering*, and *Social Sciences and Humanities*. Linking our dataset to this classification system resulted in a subset of 3,883,563 papers containing 118,012,368 citances, spanning 2000 to 2015.”

3. We have edited the text in this section to read “A variety of approaches can be used to generate these terms, and our approach is not dependent on any particular strategy. Here, we create a preliminary set of signal terms through an intensive iterative process of manually identifying, classifying, validating, and deliberating on strategies for identifying instances of disagreement. This took place over several meetings, utilizing multiple strategies to generate signal words, including sourcing cues used in related work (e.g., Bertin et al., 2016; Chen et al., 2018), expanding this list with synonyms from online thesauruses, and ranking them by their frequency among citation sentences. This inductive process included several rounds of deliberation, manual annotation, and tests of inter-rater reliability in order to generate a robust list of candidate signal terms. The terms are intended to have high validity, but are not considered comprehensive.”. In addition to this clarification, we note that a key feature of our analysis involves *evaluating* each generated cue word, and setting a specific threshold for their inclusion on our analysis, which ensures a high-degree of precision in our results.

4. We have adjusted the line to read “we also queried for citances containing both the signal terms along with at least one of four sets of filter terms, with no more than four words separating signal and filter.”

5. We have adjusted the line to read “From each set of results returned by the 65 queries, we selected 50 sentences for validation using simple random sampling without replacement (only 40 citances existed for “no consensus” +”ideas”), resulting in over 3,000 queried sentences.” which makes it clear that we are sampling from all queried results.

3) The authors have cleverly shifted the issue of negative citation to the broader concept of disagreement. They have analyzed a corpus of sentences that all contain a citation, and their aim is to capture both paper-level and community-level disagreement. However, the example of community-level disagreement given Line 188 is not necessarily an instance of disagreement between the cited works/authors, as the authors state it. It might be (it is, actually, most of the times) a list of works that do mention the existence of a debate/controversy about a topic. I think it is a typical example of a citing author agreeing with the cited authors that there is disagreement on a topic ( = agreeing on disagreement). Choosing sentences with citations and cue terms marking disagreement does not in any way guarantee that the authors are expressing disagreement with the works they cite, but merely that there is disagreement in the literature on a topic at a given time. One may therefore ask what is the point of restricting the analysis to sentences with citations and not addressing the problem from the point of view of citation polarity or function, especially if the aim is to group together paper-level and community-level disagreement. I think the citations have the same function here as filter terms, but cannot be seen as the targets of the disagreement, as they are indeed line 373 and in some sections of the Supporting Information part.

Thank you for your feedback. We attempted to summarize your points as follows, and respond to each of them in turn:

1. The example provided for *community disagreement* is not a valid instance of disagreement, as the authors have stated it

2. That “agreeing on disagreeing” can confound the author’s analysis

3. Given (1) and (2), that there is no reason to restrict to citation sentences only

4. That this presents challenges for understanding citations as *targets* of disagreement, as presented in the results and in the supporting information

1) The example provided for *community disagreement* is not a valid instance of disagreement, as the authors have stated it

The goal of our paper is not strictly to identify instances of targeted disagreement, but rather to assess, to a reasonable extent, the total amount of disagreement across scientific fields, based on traces within the scientific literature. *Paper-level* disagreements, represent direct, targeted disagreement between a citing and a cited author, which is what most people may envision when they conceive of scientific disagreements.

However, in our early manual analysis of citation data, we recognized the common case in which articles were rhetorically positioned as being in disagreement with one another by a citing author, even if the articles themselves did not explicitly disagree with one another. This kind of *community disagreement* is an indirect trace of disagreement within a field. Consider the toy example used in our paper:

“There remains controversy in the scientific literature over whether or not coffee is associated with an increased risk of cancer [1,2].”

In this case, both references [1] and [2] are *in disagreement* with one another, though they may not have even cited one another. However, the citing author has positioned them as being in disagreement, and thus indicates the presence of controversy in their research topic.

Our definition of disagreement can of course be expanded in future work, as well as differentiation made between the categories. However, we argue that community disagreement, of the type shown in our example, operates as a valid instance of disagreement, as we define it in our paper. We have edited our definition of community-level disagreement, as follows, in an attempt to clarify our position,

“Community-level disagreement, in contrast, refers to the situation in which a citing publication, without explicitly disagreeing with a cited publication, instead draws attention to a controversy or lack of consensus in the larger body of literature. Including community-level disagreement allows us to identify indirect traces of disagreement in a field, even in the absence of explicit disagreement between the referenced authors, or between the citing and cited papers.”

2) That “agreeing on disagreeing” can confound the author’s analysis

You offer an important concern, however we also argue that this fits within our notion of disagreement, as we define it in our manuscript. To demonstrate, we have slightly modified our toy example of community disagreement, as follows,

“A recent review of studies assessing the potential link between coffee consumption and cancer risk has observed continued controversy.”

In this modified example, the sentence is explicitly citing a single review article, and aggreging with its finding concerning “controversy in the literature…”. You are correct that this is not an instance of direct, *paper-level* disagreement, as the citing author is agreeing with cited paper. However, still fits within our definition of *community-level disagreement,* as it still indicates the presence of disagreement surrounding the association of coffee and cancer.

To better clarify our definition, we have added the above example citation sentence to our definition of community-level disagreement.

(3) Given (1) and (2), that there is no reason to restrict to citation sentences only

We believe that our response to (1) and (2) in this comment demonstrates that typical cases of community disagreement involve the citing author either (a) illustrating disagreement between multiple cited papers, or (b) citing a disagreement in a field as evidenced in another paper, such as a review article. Both (a) and (b) involve the citing author making use of past literature, and so it is natural to limit to in-text citations.

However, you are correct that these traces of community disagreement may not be limited to citation sentences. For example, consider the following (entirely artificial) paragraph that highlights two different ways in which relevant disagreement information appears *outside* of a citation sentence, which I label (c) and (d) respectively.

“(c) There is controversy over the link between tea and cancer, partially due to a lack of studies in the area. (d) The link between coffee and cancer, in contrast, has received more intensive study. The findings of these studies, however, are often contradictory and conflicting. “

Here, (c) demonstrates the case where controversy is remarked upon without any citations. We agree with you, that these could be legitimate instances of disagreement that are not picked up by our method. However, we maintain that valid instances of disagreement are far more likely to occur within a citation context than without. Viewed in this way, we agree that citations function similar to a filter term, providing more precise instances of disagreement. Future work should build upon our results utilizing *all* the text to identify instances of community disagreement. However, in our present study we argue that our emphasis of the non-exhaustiveness of our method, focusing instead on precision and generating a reasonable “first estimation” of disagreement. Towards this goal, restricting to only those sentences containing citations is reasonable.

In contrast, the two sentences labeled by (d) demonstrate the possibility that surrounding sentences could change the meaning of the citation sentence, signaling disagreement and controversy that is not apparent in the citance alone. We chose to limit our analysis to only a single citation sentence to simplify annotation and analysis. Early on in this project, we did assess the potential of the sentences immediately before and following the citation sentence, however we found that doing so introduced technical and conceptual challenges. We also did not find these wider citation contexts to benefit classification of disagreement to a significant extent. More sophisticated tools, such as utilizing co-reference identification, could be used in the future to automatically delineate relevant citation sentences.

With these considerations, we added the following text to our limitations section,

“Given our focus on citation sentences, we are not able to identify traces of disagreements that occur without explicit reference to past literature, or those that can only be classified as disagreement with surrounding sentences as context.”

4) That this presents challenges for understanding citations as *targets* of disagreement, as presented in the results and in the supporting information

Thank you for this point. The majority of our analysis concerns only assessing the extent of disagreement in different fields of science. However, you are correct that our notion of *community-level* disagreement confounds our analysis of *issued* and *received* disagreement citations, and section S6 of our supporting information.

It is an acknowledged limitation of our work that we are not able to discriminate between *paper-level* and *community-level* disagreement. We do not believe that this limitation significantly impacts our main results. In our discussion of the papers that most issued or received disagreement citations, we find most results, and their citation sentences, to be reasonable, apart from noted methodological artifacts.

In light of this limitation, we have made the following changes to our manuscript:

– In our Results section, we have rephrased the paragraph discussing the cited paper perspective to state “Considering the cited paper perspective—those papers that received the most paper-level disagreement citations or were referenced the most in the context of community disagreement”, which draws attention to the difference in disagreement type when interpreting these results.

– Throughout the paper, we have replaced “received disagreement citations” with the phrase “cited in the context of disagreement”, which is a phrase more inclusive of both paper and community-level disagreement.

– We have added text to the end of section S6 in our supporting information, highlighting the potential for community-level disagreement to confound results of the citation analysis, and noting the need for future research to disentangle these definitions.

– In our conclusion section, we have edited the second-to-last paragraph to read “The general method of generating and manually validating signal and filter terms can also be applied to other scientific phenomena, such as detecting uncertainty (Chen et al., 2018), negativity (Catalini et al., 2015), discovery (Small et al., 2017), or an expanded framework of disagreement (e.g., Moravcsik and Murugesan, 1975; Teufel et al., 2006).”. This mention of an “expanded framework of disagreement” is intended to draw attention to the need for further work to not only disentangle paper- and community-level citation, but also create a more expanded taxonomy of disagreement, all types of which may be relevant to understanding how and why controversy and consensus manifest in science.

4) I understand that precision is the priority in this study. Nevertheless, some equally useful vocabularies are not included in the list in Table 2. For example, commonly used terms that are missing from the list for 'studies' include research, investigation, inquiry, just to name a few. A more extensive list would certainly improve the hit rate. Similarly, some important omissions from the list for 'ideas' include concept and claim.Please either redo this part of the analysis with more terms, or discuss how this is a limitation of the study.

Thank you for this suggestion. We have added the following line into our limitation section to highlight the incompleteness of the list, and also the potential for future refinement:

“Our lists of signal and filter terms are also non-exhaustive, and so their extension in future research would identify more instance of disagreement.”

5) Related to point 4, a quick review of the available dataset and the limitations provided by the authors show that the precision may not be good either. The authors claim that false positives are marginal. They may be right. But when it comes to making analyses at the journal level, this can distort the result somewhat. I will take just one example for lack of space: Figure 3, Soc and Hum, Electoral studies (which is emphasized in the figure) -> with a manual check, I identify 37% of the citations as false positives for this journal. The use of syntactic parsing could have filtered out the false positives from all the examples citing "presidential debates" for example. The problem is that we don't know the amount of false positives at all.

We appreciate your bringing this to our attention. We agree that while our method is precise at the aggregate level, certain field-level results are confounded by false-positives. We acknowledge this as a limitation, and provide an extensive discussion of such disciplinary artifacts in the supporting information. We also argue that a unique advantage of our method, compared to black-box methods commonly used in citance classification, is that phrases like “presidential debates”, can be trivially excluded, mitigating common artifacts in sub-topics. This task is saved for future research, so that our empirical results reflect the exact queries that were manually validated.

To further emphasize this point, we have included the following line in our Results section, at the end of our paragraph introducing the meso-level field analysis

“Field-level results must be interested cautiously, however, as our signal terms may mis-classify citances based on disciplinary keywords and jargon (see Supporting Information).”

6) Is there an underlying principle that leads to the distinction between the paper- and community-level disagreements?

Thank you for this question. We introduced these concepts of paper-level and community-level disagreement following our extensive annotation and deliberation of citation data. In our observations, we found that a strict focus on paper-level disagreement, alone, was excluding many valuable traces of disagreement in the literature, that is, those cases of *community-level* disagreement in which an author highlighted and brought attention to an existing disagreement in their field. We have altered the text in the section “Operationalizing disagreement” to use the phrase “We *introduce* an inclusive definition of disagreement…”, to make it clear that this distinction was our own contribution.

While we do not do so here, we encourage future work to distinguish between paper- and community-level disagreement in their analyses. Based on our experience with these data, *community-level* disagreement is most often associated with the *+studies* filter term, which we find over-represented among the Biomedical and Health Sciences. (Figure SI 2) Further investigation may reveal other notable trends.

7) Line 260: Percent valid is defined as the percentage of citances labeled as valid by both coders.This seems very subjective. If different individuals were chosen as the coders, then we may end up with possibly quite different results because it is quite conceivable that we can find another pair of coders who may make different judgements. Are there alternatives to make this part of the procedure more systematic and reproducible?

Thank you for pointing this out. You are correct that this procedure has the potential to introduce subjectivity that could confound results. However we note that our annotation criteria was developed collaboratively among all of the authors on our manuscript, involving extensive deliberation and an iterative process of smaller-scale validation, allowing for a high degree of agreement among coders. This is evidenced in the high degree of % agreement (85.5 percent) and Cohen’s kappa (0.66) among the coders.

Moreover, for each query, a different combination of coders was randomly selected from among the authors on this paper, such that the effect of any particular combination was mitigated. We edited the first paragraph of the “Query Validation” subsection, to better clarify this process:

“For each query, two coders were randomly selected from among the seven authors on this paper to manually annotate each citance as a valid or invalid instance of disagreement”

The study of disagreement in science often necessitates a degree of manual annotation. Ideally, we could recruit and train a greater number of coders, such that each citance could be judged by more than two people. However, given the intensive labor required in this manual annotation, such an ideal is beyond the reach of our team. Still, we maintain that we followed social science best practices to mitigate such biases, including (1) carefully deliberating over our forming a consensus over annotation criteria; (2) mitigating potential subjectivity by randomly assigning combinations of coders to each query, and (3) calculating standard measures of inter-rater reliability.

We do welcome others to take advantage of our data, including the large dataset of manually-annotated citation sentences that has been made available alongside this manuscript. Readers are encouraged to review our own team’s judgements, or extend their scope.

8) Line 402-403: prioritize precision … relatively rare.If the methodology prioritizes precision, then your results cannot be used to support a conclusion that it is relatively rare because one of the many possible consequences of your priority choice is to lower the rate of appearances of qualified instances. If you expand the signal terms and filter terms, then the rate is likely to increase.

Thank you for pointing this out. We agree that our evidence does not support this claim, and so have removed this line from our Discussion section.

9) Line 420: a field is increasingly becoming consensual … (Smolin, 2007).This interpretation is not particularly convincing. A consensual field is a dying field unless it finds new driving forces. It is essential to maintain the healthy growth of a field as well as to the career and social dynamics of individual researchers who choose the work in the field. On the other hand, consensus at lower levels such as topic levels can be reached without fundamentally damaging the growth of a field. I wonder what you would see if you normalize instances by article and/or by all citances (including non-disagreement citances).

Thank you for this valuable point. There are many potential interpretations of this finding. However, we also believe that the interpretation we provide is valid, and rests on discussions and concerns provided by authors within the Physics community. We have clarified this section in an attempt to both hedge and expand upon this interpretation,

“Changes in these structural characteristics may also contribute to the temporal evolution of disagreement. For instance, the decrease of disagreement in physics and engineering may be due to a transition into a period of “normal” science (Kuhn, 1996), as it has been previously argued for certain sub-fields (Smolin, 2007). Increase in collaboration (Wuchty et al., 2007) may also affect the trends, as consensus has to be reached among a larger body of individuals during the research process.”

Our results predominantly report percentage of disagreement citances over all citances, including non-disagreement citances, which is already normalized. Exceptions are Figure 3 and Figure SI 2, which report observed/expected ratios of the percentage of disagreement per meso-level field or per query, which is in turn based on the earlier (normalized) percentage of disagreement.

10) Line 474. The framework needs more clarification. For example, how is it related to previous works on negations and uncertainties? For example, manually crafted signal terms vs. black-box approaches, especially for potentially subsequent expansions, coder-based validity vs information theoretic approaches, citer-citee disagreement vs incommensurable paradigms (one may never cite the other from an incommensurable paradigm), and the prevalence of disagreements expressed outside citances.

Thank you, we welcome the chance to further clarify our framework. We have edited the closing paragraph of our literature review to read as follows, which we believe helps to clarify the novelty of our work compared to related studies,

“Building on these studies, we propose a novel approach for the study of disagreement based on a set of manually-validated cue-phrases. We conduct one of the first empirical investigations into the specific notion of *disagreement* in science, and our inclusive definition allows us to capture both explicit disagreement between papers, as well as traces of disagreement in a field. Our cue-phrase based approach is also more transparent and reproducible than black-box machine learning methodologies commonly employed in citation classification, and also extensively validated using over 3,000 citation sentences representing a range of fields. We also extend the scale of past analyses, identifying instances of disagreement across more than four million scientific publications. “

Additionally, we reflect on the possibility that we miss non-explicit disagreements (i.e, disagreement signaled by the lack of citation between papers) in our response to essential revision 11, below.

11) Please revise the manuscript at appropriate places to address the following points:i) the study was restricted to English-language articles.ii) the absence of a citation can be an even stronger marker of disagreement than a negative citation.iii) disagreements in science can take place in other venues, such as conferences, and in other article types (such as book reviews).

Thank you for these points. In regards to the point (i), our Methods section clarifies that the documents we considered were English-language; additionally, we have clarified this in our abstract.

In regards to points (ii) and (iii), we have included the following line as part of our first limitation: